# CRISPR/Cas12a technology combined with immunochromatographic strips for portable detection of African swine fever virus

Xinjie Wang[1,8], Pinpin Ji[2,8], Huiying Fan[3,8], Lu Dang[4,8], Wenwei Wan[5], Siyuan Liu[5], Yanhua Li[6], Wenxia Yu[5], Xiangyang Li[5], Xiaodong Ma [1], Xu Ma[7], Qin Zhao[2✉], Xingxu Huang[5✉] & Ming Liao[3✉]

African swine fever virus (ASFV), the aetiological agent of African swine fever (ASF), causes lethal haemorrhagic fever in domestic pigs with high mortality and morbidity and has devastating consequences on the global swine industry. On-site rapid and sensitive detection of ASFV is key to the timely implementation of control. In this study, we developed a rapid, sensitive and instrument-free ASFV detection method based on CRISPR/Cas12a technology and lateral flow detection (named CRISPR/Cas12a-LFD). The limit of detection of CRISPR/Cas12a-LFD is 20 copies of ASFV genomic DNA per reaction, and the detection process can be completed in an hour. The assay showed no cross-reactivity with other swine DNA viruses, and has 100% agreement with real-time PCR detection of ASFV in 149 clinical samples. Overall, the CRISPR/Cas12a-LFD method provides a novel alternative for the portable, simple, sensitive, and specific detection of ASFV and may contribute to the prevention and control of ASF outbreaks.

[1] Institute for Brain Research and Rehabilitation, Guangdong Key Laboratory of Mental Health and Cognitive Science, Center for Studies of Psychological Application, South China Normal University, Guangzhou 510631, China. [2] Department of Preventive Veterinary Medicine, College of Veterinary Medicine, Northwest A&F University, Yangling 712100, China. [3] Animal Infectious Diseases Laboratory, College of Veterinary Medicine, South China Agricultural University, Guangzhou 510631, China. [4] Affiliated Cancer Hospital & Institute of Guangzhou Medical University, 78 Hengzhigang Road, Guangzhou 510095, China. [5] School of Life Science and Technology, ShanghaiTech University, 100 Haike Rd., Pudong New Area, Shanghai 201210, China. [6] College of Veterinary Medicine, Yangzhou University, Yangzhou 225009, China. [7] National Research Institute for Family Planning, No. 12 Dahuishi Rd, Beijing 100081, China.. [8] These authors contributed equally: Xinjie Wang, Pinpin Ji, Huiying Fan, Lu Dang. ✉email: qinzhao_2004@nwsuaf.edu.cn; huangxx@shanghaitech.edu.cn; mliao@scau.edu.cn

African swine fever (ASF) is a severe disease of domestic and wild swine, with a high mortality rate up to 100%[1]. The disease is caused by African swine fever virus (ASFV), which belongs to the genus *Asfivirus* within the *Asfarviridae* family[2]. ASFV is a large enveloped double-stranded DNA virus that can be transmitted by soft ticks (*Ornithodoros* genus)[3]. Based on the conserved gene *B646L* encoding the viral protein p72, ASFV is currently classified into 24 genotypes[4].

ASF was first identified in 1920 in Kenya[5]. The recent outbreak is the third ASFV incursion since 2007, when it spread to the Caucasus and Eastern Europe, resulting in outbreaks in the Russian Federation and neighbouring countries. ASF was first reported in China in August 2018 and rapidly spread, with 151 confirmed outbreaks that caused the death or culling of more than 1 million pigs by August 2019 (http://www.oie.int). To date, ASFV has spread quickly through Asia, with confirmed cases in Mongolia, Laos, Cambodia, Vietnam, and North Korea (Food and Agriculture Organization of Animal Health, 2019) making the control of its spread even more difficult. Therefore, ASFV poses a serious threat to the global swine industry[6].

ASF normally presents with high fever, cyanosis of the skin and severe haemorrhages in the lymph nodes. ASF is indistinguishable from classical swine fever (CSF) by clinical signs or post-mortem necropsy[7,8]. ASF control is mainly by animal slaughter and strict sanitary measures due to the absence of an effective vaccine or treatment, which causes major economic losses[1]. A highly sensitive and specific diagnostic method is needed for early virus detection, as this aids in the rapid implementation of control measures necessary for disease control and eradication[9]. Molecular detection of the ASFV genome has been applied in ASF diagnosis. The recommended diagnostic methods include polymerase chain reaction (PCR) and real-time PCR for ASFV genomic DNA[10,11]. However, these methods require an expensive instrument and professional operation, which limits their application in the field. Both loop-mediated isothermal amplification (LAMP) and cross-priming amplification (CPA) are also available for ASFV detection[12,13]. Although both tests are sensitive and fast, they might generate a small percentage of false-positive results[13,14]. Recently, an isothermal recombinase polymerase amplification (RPA) assay for ASFV detection was also developed[15]. However, these tests are not recommended for the detection of recovered or virus carrier animals, since their sensitivities are much lower than those of real-time PCR[16].

Recently, clustered regularly interspaced short palindromic repeats/CRISPR-associated (CRISPR/Cas) based nucleic acid detection technology was developed with the advantages of rapidity, simplicity and a low cost. The detection relies on the target-activated nonspecific endonuclease activity of Cas13 or Cas12 after binding to a specific target RNA or DNA via programmable guide RNAs[17,18]. By combining the programmable specificity of Cas12/13 with a reporter molecule that is activated upon target recognition, these enzymes result in specific and sensitive indications of the presence or quantity of nucleic acid. CRISPR/Cas-based diagnostic technology has been successfully applied to detect a variety of highly pathogenic viruses, such as Zika virus (ZIKV), Dengue virus (DENV), human papillomavirus (HPV), and Avian Influenza A Virus (H7N9)[18–20].

In this study, we developed a molecular detection system targeting the *B646L* gene for ASFV detection that combines CRISPR/Cas12a technology and the lateral flow detection method. This detection method takes advantage of the sensitivity and specificity of CRISPR/Cas12a technology and the rapid and easy result readout characteristic of a lateral flow test strip.

## Results

**CRISPR/Cas12a-based ASFV genomic DNA detection.** To sensitively and specifically detect ASFV genomic DNA, a CRISPR/Cas12a-based ASFV detection system with the LbCas12a protein, ASFV-specific CRISPR RNAs (crRNAs) and a single-stranded DNA (ssDNA) reporter (sequence TTATT) was generated as shown in Fig. 1a. The LbCas12a protein was expressed in an *Escherichia coli* system and purified properly with Ni-NTA resin (Supplementary Fig. 1). As CRISPR/Cas12a recognizes a T (thymine) nucleotide-rich protospacer-adjacent motif (PAM), ten crRNAs with a TTTN PAM targeting the *B646L* gene of genotype II ASFV were designed (Supplementary Fig. 2a). The sequence alignment result of the ASFV *B646L* gene indicated that the regions targeted by crRNA1 and crRNA6 were highly conserved in all genotypes, while the regions targeted by the remaining eight crRNAs contained single nucleotide polymorphisms (SNPs) in different genotypes (Supplementary Fig. 2b). The ssDNA reporters labelled with a quenched fluorescent molecule or with digoxin and biotin were used for fluorescent and lateral flow detection, respectively. After being activated by specific ASFV genomic DNA, CRISPR/Cas12a cleaves the ssDNA reporter molecule through its nonspecific endonuclease activity, which leads to the generation of fluorescence signals or detection via the lateral flow assay. And the fluorescence value or density of the lateral flow strip band correlates with the amount of ASFV genomic DNA.

**Detection of ASFV DNA with CRISPR/Cas12a detector.** To test whether the crRNAs could enable CRISPR/Cas12a to specifically detect ASFV DNA, ten crRNAs were tested individually and mixed (crRNAmix). CRISPR/Cas12a fluorescence detection (CRISPR/Cas12a-FD) results indicated that crRNA1 to crRNA10 and crRNAmix all highly specifically reacted with ASFV DNA, and the crRNAmix mediated a stronger fluorescence signal (Fig. 1b and Supplementary Fig. 3). A time-course study of the crRNAmix reaction with ASFV DNA was conducted to optimize the detection time, and 30 min was selected as the optimized time (Fig. 1c). Moreover, all crRNAs showed no cross-reactivity with other tested swine viruses, including double-stranded DNA virus PRV and single-stranded DNA virus PCV2 (Fig. 1c and Supplementary Fig. 4). The limit of detection of the CRISPR/Cas12a direct fluorescence readout was $2^8$ copies per reaction (Fig. 1d and Supplementary Fig. 5). To improve the detection sensitivity, we combined CRISPR/Cas12a detection and recombinase-aided amplification (RAA). As expected, the limit of detection was improved to two copies of ASFV DNA per reaction (Fig. 1e).

**Detection of the ASFV *B646L* gene with CRISPR/Cas12a lateral flow detection (CRISPR/Cas12a-LFD).** To enable the on-site diagnosis of ASFV, we designed a lateral flow readout based on the destruction of a digoxin and biotin-labelled ssDNA reporter, which allows the presentation of results with a lateral flow strip (Fig. 2a). Lateral flow detection uses nanoparticles for highly sensitive and specific signal amplification, with the advantages of simplicity, a rapid analysis, a low cost and ease of use with no requirement for skilled technicians[21]. In addition to its low cost, this method is fast and can be operated without sophisticated instruments and skilled technicians. In our lateral flow strip assay, the mouse anti-digoxin antibody conjugated gold nanoparticles specifically bind to digoxin and show the band signal on the strip. On the control line, the abundant streptavidin can bind and capture biotin-labelled ssDNA, and only the CRISPR/Cas12a-cleaved ssDNA reporter can flow through the control line. Rabbit anti-mouse antibodies on the test line bind the cleaved ssDNA with digoxin-gold nanoparticles, and the

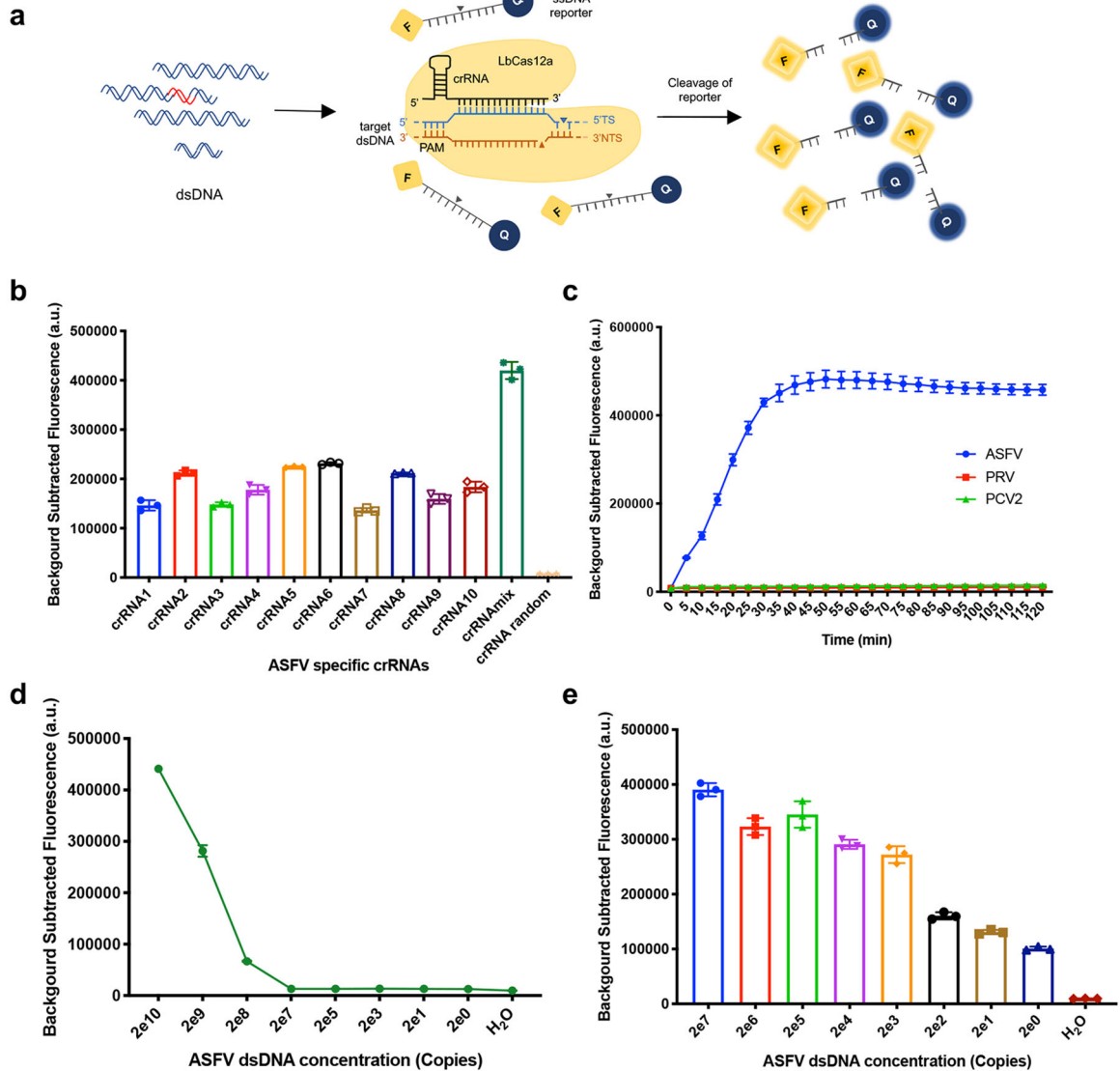

**Fig. 1 Specificity and sensitivity of ASFV genome detection with the CRISPR/Cas12a detector. a** Schematic diagram of Cas12a-mediated ASFV genome detection. Specific CRISPR RNAs (crRNAs) targeting the ASFV *B646L* gene were designed for ASFV genome detection. When CRISPR/Cas12a proteins cleave double-stranded DNA (dsDNA) with the specific crRNA guide, they induce robust, nonspecific single-stranded DNA (ssDNA) trans-cleavage. The ssDNA (sequence TTATT) reporter fluoresces when the quenched fluorescent ssDNA is cleaved. F, fluorophore; Q, quencher. **b** ASFV-specific crRNAs (crRNA1 to crRNA10) induced the strong reactivity of CRISPR/Cas12a with ASFV DNA. crRNAmix (mixed crRNA1 to crRNA10) showed a stronger signal than individual crRNAs using the same protocol. **c** The dynamics of the fluorescent signal of ASFV DNA detection with crRNAmix over a 2-h time course. The pseudorabies virus (PRV) and porcine circovirus type 2 (PCV2) were not detected with crRNAmix. **d** The sensitivity of ASFV detection with the CRISPR/Cas12a fluorescent assay in 30 min. Serially diluted synthetic ASFV DNA was used as a template. **e** The detection sensitivity of CRISPR/Cas12a combined with recombinase-aided amplification (RAA). Data points represent biologically independent replicates from at least three independent experiments, and the error bars indicate the mean ± SD.

colour change at the test line indicates a positive result (Fig. 2a). This CRISPR/Cas12a-LFD method showed great specificity, as indicated by the cross-reactivity test with other tested swine viruses (Fig. 2b, c). The limit of detection of this CRISPR/Cas12a-LFD was approximately $2^5$ copies ASFV DNA per reaction and could be improved to be 20 copies with RAA (Fig. 2d, e, Supplementary Fig. 6).

**Detection of ASFV genomic DNA with the CRISPR/Cas12a-LFD Assay**. To assess whether CRISPR/Cas12a-LFD can detect ASFV genomic DNA in complex mixtures, DNA extracted from 18 pig blood and tissue samples were used for ASFV detection by a TaqMan real-time PCR and CRISPR/Cas12a-LFD assay

(Fig. 3a). ASFV-positive results were obtained for three blood samples (CB1 to CB3) and six tissue samples (C1 to C6), which were indicated by the strong colour signal at the test line on the immunochromatographic strips (Fig. 3c, e). The signal at the test line was further quantified, and an obvious difference between the positive and negative results was observed (Fig. 3d, f). In line with the results of CRISPR/Cas12a-LFD detection, ASFV was also detected in those positive samples by both real-time PCR and CRISPR/Cas12a fluorescence detection assay (Fig. 3b, Supplementary Fig. 7, Supplementary Fig. 8).

**Performance of ASFV CRISPR/Cas12a-LFD Assay on Clinical Samples**. To rapidly detect ASFV from the clinical samples in the

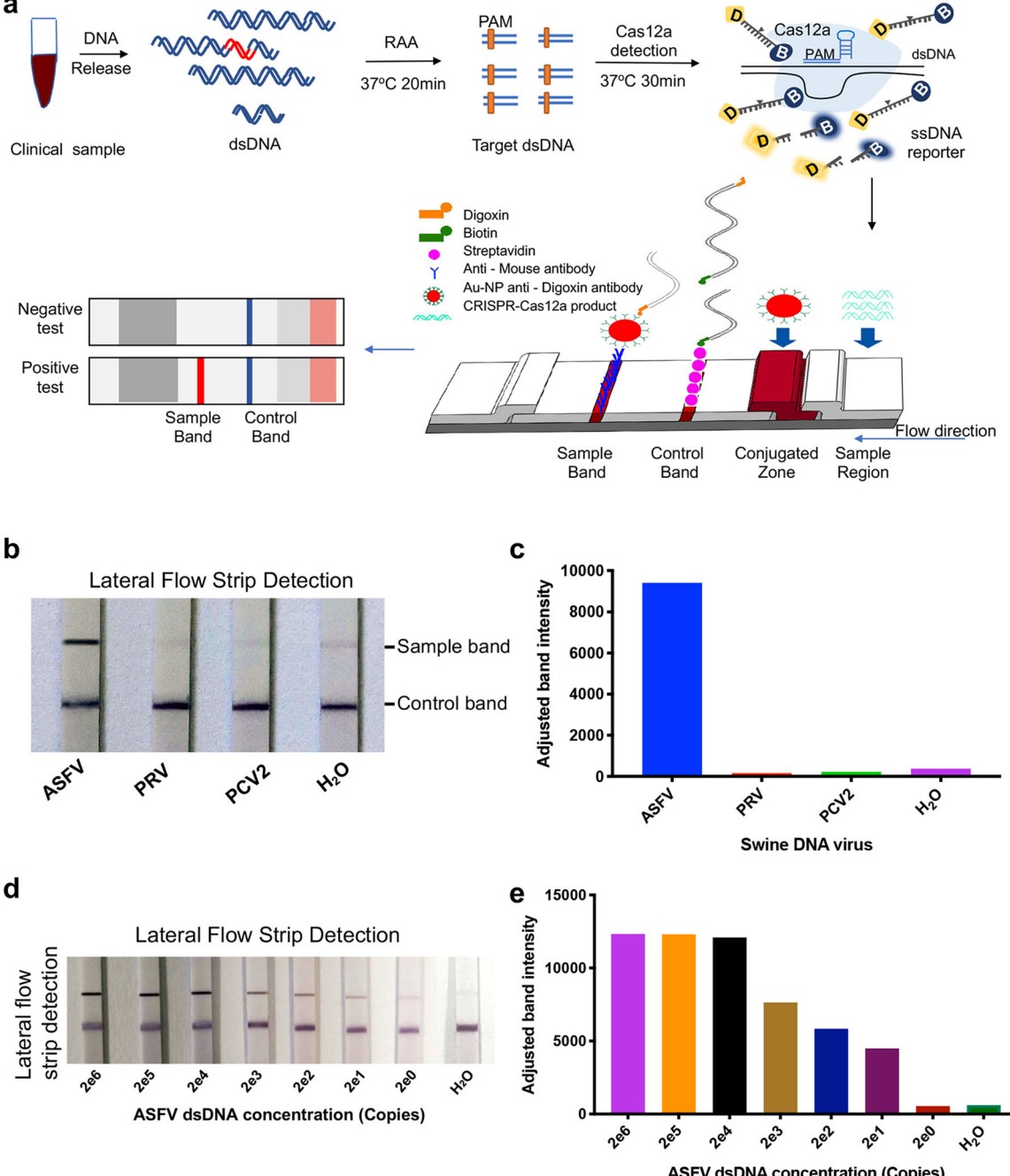

**Fig. 2 Adapting CRISPR/Cas12a for lateral flow detection (CRISPR/Cas12a-LFD). a** Schematic of ASFV detection, which combines CRISPR/Cas12a and lateral flow detection. The ssDNA reporter was labelled with digoxin and biotin (ssDNA-DB-reporter) at the 5' and 3' termini, respectively. The immunochromatographic strip using Au-NP anti-digoxin antibody to show the readout. The sample band was only shown when the ssDNA-DB-reporter was cleaved by CRISPR/Cas12a, which is activated by ASFV DNA. **b** ASFV detection with CRISPR/Cas12a-LFD at 37 °C in 30 min. The top band is the test band, and the bottom band is the control band. No colour change at the test line was observed for the two tested swine DNA viruses, PRV and PCV2. **c** The band intensity of lateral flow strip in (b) were further quantified by ImageJ and visualization with GraphPad. **d** The limit of detection of CRISPR/Cas12a-LFD combined with RAA. Serially diluted synthetic ASFV DNA was used as a template. **e** The visualization of sample bands intensity were quantified by ImageJ based on data in (d).

field, we further optimized the CRISPR/Cas12a-LFD assay. Through this optimization, the entire detection process was completed in an hour, including 3 min for releasing sample genomic DNA, 20 min for RAA replication, 30 min for the CRISPR/Cas12a reaction and readout with lateral flow detection in 3 min (Fig. 4a). A total of 149 pig serum samples were used for ASFV detection with the CRISPR/Cas12a-LFD assay without

DNA extraction. Among these samples, 86 serum samples were determined to be ASFV positive by the CRISPR/Cas12a-LFD assay (Fig. 4b, c, Supplementary Fig. 9). As a reference method for ASFV detection, real-time PCR was also performed in parallel with DNA extracted from those samples. In line with the results generated with the CRISPR/Cas12a-LFD assay, ASFV DNA was also detected in 86 samples by real-time PCR with $Ct$ values less

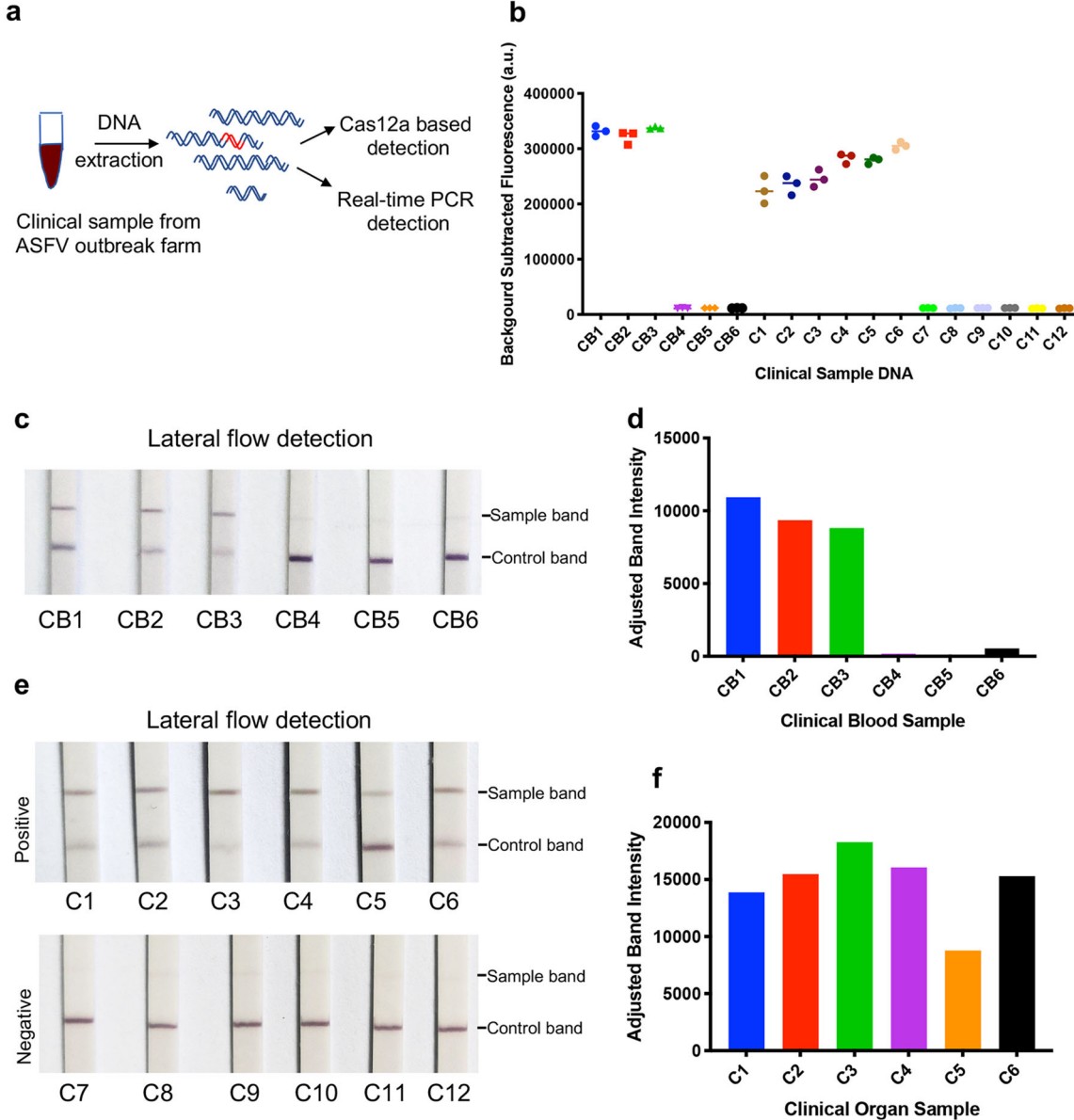

**Fig. 3 ASFV detection in clinical samples with CRISPR/Cas12a lateral flow detection. a** Schematic of ASFV detection from the clinical samples with both CRISPR/Cas12a-LFD and real-time PCR. Six blood samples (CB1 to CB6) and twelve organ samples collected from a pig farm that experienced an ASFV outbreak. The DNA was extracted and detected with both real-time PCR and CRISPR/Cas12a-LFD. **b** ASFV detection in clinical samples using CRISPR/Cas12a coupled with fluorescence readout. The fluorescence value of the negative sample was set as the background fluorescence. The detection fluorescence values were calculated by subtracting the background fluorescence value from the value of the sample detection. Samples CB1 to CB3 and C1 to C6 were positive. **c** ASFV detection in blood samples with CRISPR/Cas12a-LFD. The DNA of clinical blood samples was added to the CRISPR/Cas12a reaction system with ASFV-specific crRNAmix. After a 30 min incubation at 37 °C, the CRISPR/Cas12a reaction sample was detected with lateral flow strips at room temperature for 3 min. CB1 to CB3 were positive, which was consistent with CRISPR/Cas12a-FD and real-time PCR detection. **d** Quantitation of sample bands at the test line in (**c**). The sample bands were scanned and analysed with ImageJ software. **e** ASFV detection in organs with CRISPR/Cas12a-LFD. The detection procedure was the same as the blood DNA detection. C1 to C6 were ASFV positive, and C7 to C12 were ASFV negative. **f** Quantitation of sample bands at the test line in (**e**). The sample bands were scanned and analysed with ImageJ software.

than 36.93 (Supplementary Fig. 10). Compared with the real-time PCR, CRISPR/Cas12a-LFD also correctly identified and differentiated all 86 positive samples and showed 100% agreement with it (Fig. 4d, Table 1). There was no difference between the detection results of CRISPR/Cas12a-LFD and real-time PCR detection with ASFV. The kappa value (κ) of CRISPR/Cas12a-LFD and real-time PCR was 1.0 ($p < 0.001$). Furthermore, the quantified signals at the test line of CRISPR/Cas12a-LFD were well correlated with the $Ct$ values from the real-time PCR results (Fig. 4e).

## Discussion

ASFV infection usually leads to pig death within four days post-infection even without clinical manifestations and antibody responses. ASFV genomic DNA can be detected in blood as early as 56 h post-infection, which is two days before clinical signs emerged[16]. The endemic spread of ASFV in many countries around the world is a great threat to the global swine industry[6]. The urgent requirement for a rapid diagnosis has highlighted the necessity to improve well-validated diagnostic methods or develop new methods. Real-time PCR with high sensitivity and

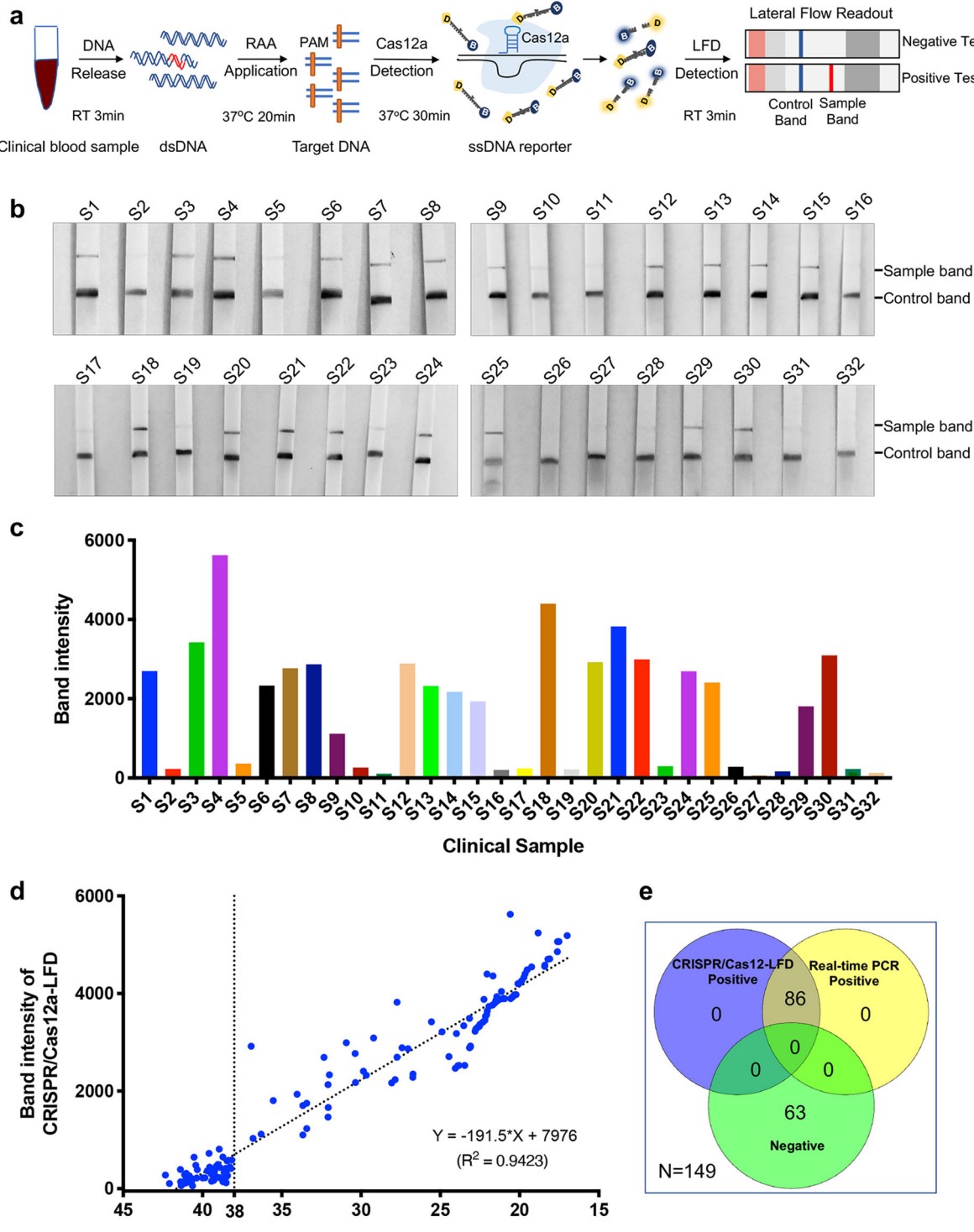

**Fig. 4 ASFV detection with CRISPR/Cas12a lateral flow readout from blood samples without DNA extraction. a** Schematic diagram of ASFV DNA detection in clinical blood samples using CRISPR/Cas12a-LFD without DNA extraction. To detect ASFV DNA in an hour, the clinical sample DNA was released in 3 min, followed by 20 min for RAA replication and 30 min for the CRISPR/Cas12a reaction; then, readout with lateral flow detection occurred in 3 min. **b** The detection of ASFV in serum samples with CRISPR/Cas12a-LFD. The DNA from clinical serum samples was released and preamplified. The CRISPR/Cas12a reaction and readout in lateral flow strips were then performed. The control band is shown at the bottom, and the test band is shown at the top. **c** Quantitation of sample bands in (b). The sample bands were scanned and analysed with ImageJ software. **d** The correlation analysis of ASFV detection results generated with CRISPR/Cas12a-LFD and real-time PCR. The same sample detection Ct value of real-time PCR and band intensity value of the lateral flow strip were co-analysed. The correlation value was analysed with GraphPad. **e** The Venn diagram shows the consistency between the Cas12a-LFD and real-time PCR assays.

**Table 1 The performance of CRISPR/Cas12a-LFD for ASFV detection in clinical samples compared with real-time PCR.**

| Methods | CRISPR/Cas12a-LFD | | | Comparison of two methods | | |
|---|---|---|---|---|---|---|
| Real-time PCR | Positive | Negative | Total | Sensitivity (%) | Specificity (%) | Kappa |
| Positive | 86 | 0 | 86 | 100 | 100 | 1.00 |
| Negative | 0 | 63 | 63 | | | |
| Total | 86 | 63 | 149 | | | |

specificity is considered the gold standard test for ASFV genome detection[16]. Since real-time PCR must be performed in a diagnostic laboratory with sophisticated equipment and professional operation, it is not suitable for on-site detection. Furthermore, diagnostic laboratories are usually far from pig farms, which may lead to a delayed result. Recently, an alternative on-site rapid ASFV detection assay, RPA with lateral flow dipstick (RPA-LFD), was developed[15]. The sensitivity of RPA-LFD is 150 copies per reaction, which is lower than that of real-time PCR. To overcome this disadvantage, we invented a new ASFV detection method that combines CRISPR/Cas12a and lateral flow readout, namely, ASFV CRISPR/Cas12a-LFD. CRISPR/Cas12a-LFD can be used for on-site ASFV detection with a sensitivity of 20 copies ASFV DNA per reaction (Fig. 3d), and the entire detection process from sampling to result takes only an hour.

To rapidly detect the ASFV genome from a clinical sample on-site, each step was optimized to reduce the detection time. Because DNA extraction was avoided, the detection time for ASFV was reduced greatly. In this study, we established a protocol for ASFV detection using DNA released from clinical samples without the classical DNA extraction step. To prepare ASFV DNA compatible with the isothermal amplification reaction, we tried to release ASFV DNA with extraction buffers from commercial kits. The pretreatment conditions were finalized by incubation with lysis buffer for 3 min followed by the addition of stabilizing buffer. Using this optimized detection protocol, we completed ASFV detection from clinical samples in an hour. Although the detection results between the two methods were 100% consistent (Fig. 4), the CRISPR/Cas12a-LFD assay requires less time than real-time PCR. Moreover, as sophisticated equipment is not required, the CRISPR/Cas12a-LFD assay will not only reduce the cost but also simplify the detection process. Therefore, the CRISPR/Cas12a-LFD assay will be a great option for on-site ASFV diagnosis.

In conclusion, we developed CRISPR/Cas12a-LFD for rapid ASF diagnosis, which can be used for the robust, specific, sensitive and portable detection of the ASFV genome. CRISPR/Cas12a-LFD has great potential for on-site ASFV detection, which may make an important contribution to ASF control.

## Methods
**Clinical samples and ethics statement**. Clinical samples used in this study were collected and treated in strict accordance with the standard operation for ASFV by the World Organization for Animal Health. ASFV suspected serum samples were collected from a farm that experienced an ASFV outbreak. All sample treatments were conducted in the African Swine Fever Regional Laboratory of China (Guangzhou). ASFV was inactivated in a BSL-3 laboratory, and viral DNA was prepared in a BLS-2 laboratory.

**Nucleic acid preparation**. The B646L gene fragment of genotype II ASFV or other genotypes was synthesized by GenScript (Nanjing, China) and cloned into the pUC57 vector. The construction of crRNA plasmids and oligonucleotides was performed by annealing and ligation into the BsaI-linearized pUC57-T7-crRNA expression vector. The sequences of oligonucleotides used for plasmid construction are listed in Supplementary Table 1.

**Genomic DNA preparation from clinical samples**. Two protocols were utilized to prepare genomic DNA from clinical samples. For the first protocol, genomic DNA

of tissue homogenate and serum samples was extracted with AxyPrep Body Fluid Viral DNA/RNA Miniprep Kit (Axygen Scientific, Silicon Valley, USA) according to the manufacturer's instructions. The other protocol was used to release DNA from serum samples without DNA extraction using the Room Temp Sample Lysis Kit (Vazyme Biotech Co., Ltd, Nanjing, China). Briefly, 2 μL of serum was mixed with 20 μL of lysis buffer, incubated at RT for 3 min, supplemented with 20 μL of stabilizing buffer, and further mixed with a vortex.

**Protein expression and purification**. The LbCas12a protein used in the ASFV detection system was expressed in an E. coli expression system. The genes encoding LbCas12a were codon-optimized and cloned into the expression vector pET-28a. The soluble LbCas12a with a TEV cleavage site and 6 × His-tag at the C-terminus was expressed and purified as described previously[18] with the following modifications. The E. coli BL21 (DE3) strain was used for protein expression with an induction of 1 mM IPTG at 16 °C for 16 h. The bacterial cells were harvested and disrupted by a high-pressure cell crusher in lysis buffer [25 mM Tris-HCl, pH 8.0, 500 mM NaCl, 10% (v/v) glycerol, 0.5 mM PMSF]. Soluble LbCas12a was purified using Ni-NTA resin and incubated with TEV protease at 4 °C overnight to remove the C-terminal His-tag. The protein was further purified by a Superdex 200 Increase filtration column (GE Healthcare Life Sciences, Connecticut, USA) via fast protein liquid chromatography (FPLC). The purified Cas12a protein was concentrated into storage buffer [50 mM Tris-HCl, pH 7.5, 500 mM NaCl, 10% (v/v) glycerol, 2 mM DTT], quantitated using the BCA Protein Assay Kit (Thermo Fisher Scientific, Massachusetts, USA), and frozen at −80 °C until use.

**crRNA preparation**. crRNA was transcribed with pUC57-crRNA as a template using the MEGAshortscript T7 Transcription Kit (Thermo Fisher Scientific, Massachusetts, USA). crRNAs were purified with the MEGAclear Kit (Thermo Fisher Scientific, Massachusetts, USA) and recovered by alcohol precipitation according to the manufacturer's instructions. RNAs were aliquoted and stored at −80 °C until use. The names and spacer sequences of different crRNAs are listed in Supplementary Table 2.

**Isothermal amplification**. The isothermal amplification of the B646L gene was performed with a commercial RAA kit (Qitian biological Co., Ltd., Jiangsu, China) according to the manufacturer's instructions. The ASFV RAA primers, namely, p72-RAA-F1 and p72-RAA-R1 (Supplementary Table 2), were designed according to the instructions. Briefly, a 50 μL reaction assembled with 2 μL DNA sample, 2 μL p72-RAA-F (forward primer, 10 μM), 2 μL p72-RAA-R (reverse primer, 10 μM), and 2.5 μL magnesium acetate (280 mM) was incubated at 39 °C for 20 min. Then, the RAA reaction was transferred to the CRISPR/Cas12a cleavage assay.

**CRISPR/Cas12a detection reaction**. Detection assays were performed with 200 ng purified LbCas12a, 25 pM ssDNA FQ probe sensor, 1 μM crRNA and 2 μL sample in a reaction buffer (100 mM NaCl, 50 mM Tris-HCl, 10 mM MgCl₂, 100 μg/mL BSA, pH 7.5) in a 20 μL reaction at 37 °C. A PerkinElmer EnSpire reader was used for fluorescence detection. Fluorescence kinetics were monitored using a monochromator with excitation at 485 nm and emission at 520 nm.

**Preparation of the rapid immunochromatographic strips**. A lateral flow detection strip was prepared using a method reported previously with minor modifications[22]. Briefly, the test strip was made of a sample pad, conjugate release pad, absorbent pad, and nitrocellulose membrane. The sample and conjugate pads were treated with blocking buffers and dried at 37 °C overnight. Streptavidin conjugate and rabbit anti-mouse IgG were used for the control and test lines, respectively, and coated to the nitrocellulose membrane. The nanogold particles were produced by the reduction of gold chloride with 1% trisodium citrate. The lateral flow strips were stored at 4 °C until use.

**Lateral flow detection reactions**. In lateral flow detection, the reporter ssDNA probe sensor was labelled with digoxin and biotin at the 5′ and 3′ termini, respectively. The CRISPR/Cas12a detection reaction was diluted 1:3 in lateral flow detection buffer, and then the strips were inserted and incubated at room temperature for 3 min. The strips were then removed and photographed using a smartphone camera. To quantify and visualize the ASFV DNA amount in the test sample, the band density was analysed with ImageJ according to ImageJ User

Guide. Briefly, the image of later flow strip is converted to grayscale and select sample band lane with rectangular selections. Then plot lanes to draw a profile plot of each lane, which represents the relative density of the target band. And then band density visualized with GraphPad Prism 7.0.

**Real-time PCR detection**. The TaqMan *real-time PCR* detection of the ASFV *B646L* gene was carried out using a Quant Studio 5 system (Applied Biosystems, Massachusetts, USA) according to the OIE-recommended procedure described previously[23] with the following modifications. Briefly, single-tube PCRs were prepared containing 10 μL of 2× AceQ qPCR Probe Master Mix (Vazyme Biotech Co., Ltd, Nanjing, China), 0.4 μL of primer F (5′-ATAGAGATACAGCTCTTC CAG-3′, 10 μM), 0.4 μL of primer R (5′-GTATGTAAGAGCTGCAGAAC-3′, 10 μM), 0.2 μL of TaqMan probe (5′-FAM-TATCGATAAGATTGAT-MGB-3′, 10 μM), 2 μL of DNA, and 7 μL of ddH$_2$O. The amplification conditions used were an initial denaturation step of 95 ℃ for 3 min, followed by 45 cycles of 95 ℃ for 15 s, 52 ℃ for 10 s, and 60 ℃ for 35 s. Fluorescence information was collected at the 60 ℃ annealing extension per cycle. The cycle value (Ct) ≤ 38.0 was judged as ASFV positive.

**Statistics and reproducibilty**. All results generated from at least three independent experiments are presented as the mean ± SD. Statistical analyses and graphing were carried out with GraphPad Prism 7.0.

**Reporting summary**. Further information on research design is available in the Nature Research Reporting Summary linked to this article.

## Data availability

All data that support the findings of this study are available from the corresponding author upon reasonable request.

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

## Acknowledgements

The authors thank the members of Huang lab, Liao lab and Zhao lab for their helpful discussions and advice. This work was supported by grants from the National Key R&D Program (2016YFA0500903, 2016YFC1000307), the National Natural Science Foundation of China (81830004), and Young Teachers Research and Development Fund from South China Normal University (18KJ06).

## Author contributions

X.H., Q.Z. and M.L. conceived and designed the project. X.W., P.J. and L.D. performed the most experiments with assistance from H.F., W.W., S.L., Y.L., W.Y. and X.L. Y.L. and L.D. analysed the data. X.W. and Y.L. wrote the manuscript with review comments from all authors. Q.Z., Xi.M. and Xu.M. edited the paper. X.H., Q.Z. and M.L. supervised and managed the project.

## Competing interests

The authors declare no competing interests.
