## [Peer Review File · Communications Biology]

Reviewers' comments:

Reviewer #1 (Remarks to the Author):

The manuscript describes a novel and highly sensitive method of ASFV DNA detection based on CRISPR/Cas12 technology. The method was successfully used for other viruses and the authors demonstrate its applicability to ASFV diagnostics. The developed CRISPR/Cas12 assay can be used for ASFV genotyping as well.

The manuscript is of interest to the readers. However several questions should be addressed to better convey the results.

Major comments:

1. The number of tested samples is below recommended. Therefore, the conclusion can not be accurate in declaring the sensitivity and specificity of the diagnostic assay. Please refer to the OIE guideline.
2. Fluorescent signal reading requires special equipment which will limit the assay usage and implementation in the diagnostic scheme.
3. ASFV genotyping based on the fluorescent signal strength doesn't look robust enough for diagnostic purposes and better be deleted. What is a cut-off (i.e. fluorescent value) to different ASFV genotypes? The manuscript should be focused on CRISPR/Cas12 diagnostic characteristics only with in-depth analysis.

Minor comments:

1. The manuscript contains some spelling errors. The authors should seek a piece of advice from a native English speaker.
2. Please provide more information in Figures captions. The text should provide a thorough description of the results depicted in the figures without referring to the text.

Reviewer #2 (Remarks to the Author):

This paper by Wang, Ji, Fan, Dang, et al. develops a Cas12-based assay for detecting African Swine Fever Virus (ASFV) with a lateral flow readout. They also demonstrate good agreement with real-time PCR assays for ASFV, and assays that can differentiate between different strains of ASFV. I like this paper overall, and think that, with minor revisions, it should be suitable for publication.

Major Comments

1. Cross-reactivity was performed against very unrelated viruses (from a sequence perspective). The authors should perform some additional tests with more closely-related viruses to verify that their test is species-specific. This can be done using synthetic targets, or viral seedstocks.
2. In Figure 3, are C7-C12 negative controls? Uninfected animals? Please clarify this in the figure, figure legend, and main text.
3. The authors should clarify how the genotyping works in Figure 5. Are multiple different reporter sequences being used? Multiple fluorophores? I am a bit confused by the presentation here.

Minor Comments

1. There are a number of grammatical errors, most notably in the abstract and introduction (but also in the figures, e.g. "copys"). The authors should do additional proofreading to address these.

2. How were the lines in Figure 4c determined? The authors should provide details about the band quantification in the Methods section, and explain how the line is determined in the figure caption. How are the band intensities adjusted?

3. The Methods section says that the RAA primer concentration used was 20 μM (Line 288). This is extremely high - is there a typo here, or some missing information?

4. As I understand it from Fig. 1c, the authors determined the optimal timing for lateral flow detection based on fluorescence kinetics from a different reporter. Is this truly the optimal time for lateral flow detection? In my experience, different reporters can lead to different kinetics - the authors should discuss this, or provide more reasoning here.

MS title: CRISPR/Cas12a Technology Combined with Immunochromatographic Strips for Portable Detection of African Swine Fever Virus

Author: Xinjie Wang, et al.,

MS number: COMMSBIO-19-1263A

A point-by-point response to the reviewer(s)' comments

We thank the reviewers for their insightful comments, which have led to a great improvement in our manuscript. The comments have been carefully taken into account and a revised submission with highlighted changes has been uploaded. Please find our responses to each comment in blue font.

Reviewers' comments:

Reviewer #1 (Remarks to the Author):

The manuscript describes a novel and highly sensitive method of ASFV DNA detection based on CRISPR/Cas12 technology. The method was successfully used for other viruses and the authors demonstrate its applicability to ASFV diagnostics. The developed CRISPR/Cas12 assay can be used for ASFV genotyping as well.

The manuscript is of interest to the readers. However several questions should be addressed to better convey the results.

Reply to the reviewer: Thank you for your favorable comments. Specific answers to each question are listed as the follows:

Major comments:

1. The number of tested samples is below recommended. Therefore, the conclusion can not be accurate in declaring the sensitivity and specificity of the diagnostic assay. Please refer to the OIE guideline.

Reply to the reviewer: Thanks for your constructive comment. To better declaring the sensitivity and specificity of CRISPR/Cas12 based ASFV detection assay, another 117 blood samples were included for ASFV detection. According to the OIE guideline, the sensitivity and specificity of our assay were calculated. The agreement of CRISPR/Cas12a-LFD and real-time PCR for ASFV detection was added to line 167~172 and Table 1 in the revised manuscript. The correlation between CRISPR/Cas12a-LFD and real-time PCR was updated in figure 4D and 4E.

“Compared with the real-time PCR, CRISPR/Cas12a-LFD also correctly identified and differentiated all 86 positive samples and showed 100% agreement with it (Fig. 4D, Table 1). There was no significant difference between the detection results of CRISPR/Cas12a-LFD and real-time PCR detection with ASFV. The kappa value (κ) of CRISPR/Cas12a-LFD and real-time PCR was 1.0 ($p < 0.001$).”

Table 1 The performance of CRISPR/Cas12a-LFD for ASFV detection in clinical samples compared with real-time PCR

Methods	CRISPR/Cas12a-LFD			Comparison of two methods		
Real-time PCR	Positive	Negative	Total	Sensitivity (%)	Specificity (%)	Kappa
Positive	86	0	86	100	100	1.00
Negative	0	63	63			
Total	86	63	149			

Figure 4D and 4E. (D) The correlation analysis of ASFV detection results generated with CRISPR/Cas12a-LFD and real-time PCR. The same sample detection Ct value of real-time PCR and band intensity value of the lateral flow strip were co-analysed. The correlation value was analysed with GraphPad. (E) The Venn diagram shows the consistency between the Cas12a-LFD and real-time PCR assays.

2. Fluorescent signal reading requires special equipment which will limit the assay usage and implementation in the diagnostic scheme.

Reply to the reviewer: Thanks for your comment. Fluorescent signal reading is not required for ASFV detection using CRISPR/Cas12a-LFD. So, we don't think the fluorescent signal detection will not limit the on-site ASFV diagnostic with CRISPR/Cas12a-LFD.

3. ASFV genotyping based on the fluorescent signal strength doesn't look robust enough for diagnostic purposes and better be deleted. What is a cut-off (i.e. fluorescent value) to different ASFV genotypes? The manuscript should be focused on CRISPR/Cas12 diagnostic characteristics only with in-depth analysis.

Reply to the reviewer: Thanks for your constructive comment. In the previous manuscript, ASFV genotyping based on fluorescent readout needs a reader and further improvement. As your suggestion, in this revised manuscript we will delete the genotyping part and focus on the rapid, sensitive, specific ASFV detection.

Minor comments:

1. The manuscript contains some spelling errors. The authors should seek a piece of advice from a native English speaker.

Reply to the reviewer: Sorry for the spelling errors. The revised manuscript has been carefully edited with the Nature Research Editing Service (SNAS verification code: 8A8E-39AB-5D0F-6BC1-00FE).

2. Please provide more information in Figures captions. The text should provide a thorough description of the results depicted in the figures without referring to the text.

Reply to the reviewer: Thank you very much for your suggestions. In the revised manuscript, we rephrased the figure legends.

Reviewer #2 (Remarks to the Author):

This paper by Wang, Ji, Fan, Dang, et al. develops a Cas12-based assay for detecting African Swine Fever Virus (ASFV) with a lateral flow readout. They also demonstrate good agreement with real-time PCR assays for ASFV, and assays that can differentiate between different strains of ASFV. I like this paper overall, and think that, with minor revisions, it should be suitable for publication.

Major Comments

1. Cross-reactivity was performed against very unrelated viruses (from a sequence perspective). The authors should perform some additional tests with more closely-related viruses to verify that their test is species-specific. This can be done using synthetic targets, or viral seedstocks.

Reply to the reviewer: Thank you very much for your suggestions. We chose two common swine DNA viruses for the cross-reactivity validation, including Pseudorabies virus (PRV) and porcine circovirus type 2 (PCV2). As shown in the manuscript, there is no cross-reactivity with both DNA viruses. We further blasted the ASFV B646L gene in the GenBank database, and no sequence over 90% similarity was found except ASFV. Furthermore, the CRISPR/Cas based detection had been reported is highly specificity and sensitivity with the mismatch between crRNA and target sequence^{1,2}. Taken together, we didn't find any cross-reactivity of the CRISPR/Cas12a based ASFV detection.

2. In Figure 3, are C7-C12 negative controls? Uninfected animals? Please clarify this in the figure, figure legend, and main text.

Reply to the reviewer: Yes, the C7 to C12 are negative samples from uninfected pigs determined as ASFV-negative by an ASFV real-time PCR assay. We have clarified it clearly in the revised manuscript.

3. The authors should clarify how the genotyping works in Figure 5. Are multiple different reporter sequences being used? Multiple fluorophores? I am a bit confused by the presentation here.

Reply to the reviewer: Thanks for your constructive comment. In the previous manuscript, the ASFV genotyping based on the fluorescent signal need a reader and more modification to improve the readout signals. Based on the reviewer's comments, in this revised paper we deleted the genotyping part to focus on specific ASFV detection.

Minor Comments

1. There are a number of grammatical errors, most notably in the abstract and introduction (but also in the figures, e.g. "copys"). The authors should do additional proofreading to address these.

Reply to the reviewer: Sorry for the grammatical and spelling errors. The revised manuscript has been carefully edited with the Nature Research Editing Service (SNAS verification code: 8A8E-39AB-5D0F-6BC1-00FE).

2. How were the lines in Figure 4c determined? The authors should provide details about the band quantification in the Methods section, and explain how the line is determined in the figure caption. How are the band intensities adjusted?

Reply to the reviewer: Thanks for your constructive comment. More information about the band quantification analyze was added in the revised manuscript. To quantify and visualize the ASFV DNA amount in the tested sample, the band density was analysed with ImageJ according to the ImageJ User Guide, and have been added to line 295~299 in the revised manuscript. The band intensity value shown in the graph is based on the ImageJ calculate value and no adjustment was performed. We have corrected the "Adjusted band intensity" to "Band intensity" in the revised manuscript.

"To quantify and visualize the ASFV DNA amount in the test sample, the band density was analysed with ImageJ according to ImageJ User Guide. Briefly, the image of later flow strip is converted to grayscale and select sample band lane with rectangular selections. Then plot lanes to draw a profile plot of each lane, which represents the relative density of the target band. And then band density visualized with GraphPad Prism 7.0."

3. The Methods section says that the RAA primer concentration used was 20 μM (Line 288). This is extremely high - is there a typo here, or some missing information?

Reply to the reviewer: Thanks for your comment. It is a typo mistake and has been corrected in the revised manuscript. In a 50 μL reaction of RAA amplification, 2 μL forward primer (10 μM) and 2 μL reverse primer (10 μM) were used according to the manufacturer's instructions (Qitian biological Co., Ltd., Jiangsu, China).

4. As I understand it from Fig. 1c, the authors determined the optimal timing for lateral flow detection based on fluorescence kinetics from a different reporter. Is this truly the optimal time for lateral flow detection? In my experience, different reporters can lead to different kinetics - the authors should discuss this, or provide more reasoning here.

Reply to the reviewer: Thanks for your comment. The CRISPR/Cas12a mediate nonspecific endonuclease activity when activated by the target DNA sequence, and the cleavage of the reporter will give the readout signal. The reaction plateau will reach when all the input ssDNA reporters were cleavage. An identical sequence of ssDNA reporter was used in fluorescence detection (ssDNA-FQ) and lateral flow detection (ssDNA-BD), so the cleavage efficacy by CRISPR/Cas12a should be same³. Based on these, the detection time point of fluorescence signal detection can be applied in the CRISPR/Cas12a-LFD.

- 1 Myhrvold, C. *et al.* Field-deployable viral diagnostics using CRISPR-Cas13. *Science* **360**, 444-448, doi:10.1126/science.aas8836 (2018).
- 2 Chen, J. S. *et al.* CRISPR-Cas12a target binding unleashes indiscriminate single-stranded DNase activity. *Science* **360**, 436-439, doi:10.1126/science.aar6245 (2018).
- 3 Gootenberg, J. S. *et al.* Multiplexed and portable nucleic acid detection platform with Cas13, Cas12a, and Csm6. *Science* **360**, 439-444, doi:10.1126/science.aaq0179 (2018).

REVIEWERS' COMMENTS:

Reviewer #1 (Remarks to the Author):

The manuscript was significantly improved. All the comments and questions have been addressed. This version of the manuscript is appealing to the readers of the Communications Biology.